# Recovery from heat shock requires the microRNA pathway in *Caenorhabditis elegans*

**Delaney C. Pagliuso**[1], **Devavrat M. Bodas**[1,2], **Amy E. Pasquinelli**[1]*

**1** Division of Biology, University of California, San Diego, La Jolla, California, United States of America,
**2** Department of Molecular Biology and Genetics, Johns Hopkins University School of Medicine, Baltimore, Maryland, United States of America

* apasquinelli@ucsd.edu

**Data Availability Statement:** All RNA Sequencing data files are available from the GEO database (https://www.ncbi.nlm.nih.gov/geo/query/acc.cgi?acc=GSE168073__;!!Mih3wA!VTJBMQz4Sl5gF6hiGg8GqgFSETUZS7vzl3WyXfhPD

## Abstract

The heat shock response (HSR) is a highly conserved cellular process that promotes survival during stress. A hallmark of the HSR is the rapid induction of heat shock proteins (HSPs), such as HSP-70, by transcriptional activation. Once the stress is alleviated, HSPs return to near basal levels through incompletely understood mechanisms. Here, we show that the microRNA pathway acts during heat shock recovery in *Caenorhabditis elegans*. Depletion of the miRNA Argonaute, Argonaute Like Gene 1 (ALG-1), after an episode of heat shock resulted in decreased survival and perdurance of high *hsp-70* levels. We present evidence that regulation of *hsp-70* is dependent on miR-85 and sequences in the *hsp-70* 3'UTR that contain target sites for this miRNA. Regulation of *hsp-70* by the miRNA pathway was found to be particularly important during recovery from HS, as animals that lacked miR-85 or its target sites in the *hsp-70* 3'UTR overexpressed HSP-70 and exhibited reduced viability. In summary, our findings show that down-regulation of *hsp-70* by miR-85 after HS promotes survival, highlighting a previously unappreciated role for the miRNA pathway during recovery from stress.

## Author summary

In the natural world, organisms constantly face stressful conditions such as oxidative stress, pathogen infection, starvation and heat stress. While many studies have focused on the cellular response to stress, less is known about how gene expression re-sets after the stress has been ameliorated. Here, we show that the microRNA pathway plays a critical role during the recovery phase after an episode of heat shock in the nematode, *Caenorhabditis elegans*. Elevated temperatures induce high expression of heat shock proteins (HSPs), including HSP-70, that provide protection from the damaging effects of high heat. We found that restoration of basal levels of HSP-70 after heat shock depends on Argonaute Like Gene 1 and miR-85. Moreover, loss of miRNA-mediated repression of HSP-70 results in compromised survival following heat shock. Our study draws attention to the recovery phase of the heat shock response and highlights an important role for the microRNA pathway in re-establishing gene expression programs needed for organismal viability post stress.

YrddepB385AacGhMNa6RV1M$) with accession number GSE168073.

**Funding:** Support for this study was provided by a UCSD Cellular and Molecular Genetics Training Program institutional grant from National Institute of General Medical Sciences (https://www.nigms.nih.gov/) (T32 GM007240) (D.C.P.) and a University of California, San Diego Eureka! Scholarship (D.M.B.). This work was funded by a grant from the NIH through NIGMS (https://www.nigms.nih.gov/) (GM127012) to A.E.P. Some strains were provided by the C. elegans Genetics Center (CGC) (https://cgc.umn.edu), which is funded by NIH Office of Research Infrastructure Programs (P40 OD010440) (https://orip.nih.gov). The funders had no role in study design, data collection and analysis, decision to publish, or preparation of the manuscript.

**Competing interests:** The authors have declared that no competing interests exist.

## Introduction

The heat shock response is a broadly conserved cellular mechanism that is activated to help organisms survive stress [1]. Various types of stress including heat shock, oxidative stress, infection, and tumorigenesis can trigger the heat shock response (HSR) [2,3]. Upon encountering stress, proteins can misfold and form harmful aggregates, which have the potential to cause cell death if not ameliorated [4–6]. To maintain protein homeostasis (proteostasis), the HSR stimulates a transient yet robust reprogramming of cellular activities, coupling a general decline in transcription and translation with the specific induction of molecular chaperones known as heat shock proteins (HSPs) [7–9]. Upon stress, activation of the transcription factor HSF1 (Heat Shock Factor 1) drives the accumulation of HSPs, which aid in restoring proteostasis and are, thus, integral to organismal survival [10]. Many HSPs have a diversity of cellular functions under both stressful and non-stressful conditions that revolve around mediating protein folding, stability and complex formation [11,12]. This group includes HSP70, which is constitutively expressed in most cell types but is robustly induced by heat shock and other types of stress. While up-regulation of HSP70 upon stress is primarily a result of HSF1 dependent transcription, less is understood about how HSP70 levels return to basal levels after stress [13]. Studies in *Drosophila* and human cells have documented the rapid induction and subsequent decline of *Hsp70* transcripts during the HSR. Destabilization of *Hsp70* mRNA after return to ambient temperatures is regulated post-transcriptionally and involves yet to be defined 3'UTR sequences and trans-acting factors [14–17].

In addition to the transcriptional reprogramming of protein coding genes elicited upon HS, microRNAs (miRNAs) have also been reported to respond to HS. MiRNAs are short, ~22 nucleotide non-coding RNAs that guide Argonaute (AGO) proteins to target mRNAs, triggering destabilization through imperfect base pairing [18]. In *Caenorhabditis elegans*, ALG-1 (AGO-Like Gene 1) is broadly expressed and binds most miRNAs [19]. Specific *C. elegans* miRNAs have been observed to be up- or down-regulated during HS [20–22], suggesting roles in the HSR. Moreover, deletion of *miR-71* or *miR-246* results in reduced HS survival [23]. Despite the identification of individual miRNAs that influence HS survival, much is yet to be learned about specific roles and targets of miRNAs during stress.

In this study, we demonstrate a role for the miRNA pathway in the recovery phase following heat shock in *C. elegans*. Removal of ALG-1 after HS resulted in reduced survival and perdurance of higher *hsp-70* levels. We found that efficient down-regulation of *hsp-70* after an episode of HS also requires miR-85 and the *hsp-70* 3'UTR, which contains two miR-85 binding sites. Moreover, animals lacking miR-85 or its binding sites in the *hsp-70* 3'UTR exhibited greater sensitivity to HS. The reduced survival of these strains was dependent on *hsp-70* expression, as knock down of *hsp-70* by RNA interference (RNAi) after HS restored viability to WT levels. Altogether, we show that in *C. elegans* down-regulation of *hsp-70* after HS is facilitated by miR-85 and is important for organismal survival.

## Results

### The microRNA pathway is important for recovery after heat shock

To investigate the role of the miRNA pathway in the *C. elegans* heat shock response (HSR), we first tested if Argonaute Like Gene 1 (ALG-1) was important for survival after exposure to elevated temperatures. Populations of wild type (WT) and *alg-1(gk214)* loss-of-function mutants were subjected to 4 hrs of HS at 35˚C, and the percent survival was calculated after allowing the animals to recover for 24 hrs at 20˚C. Under these conditions, about 60% of the WT and only 30% of the *alg-1(gk214)* animals survived (Fig 1A). Loss of *alg-2*, the most closely related

homolog to *alg-1* in *C. elegans* [24], did not affect survival in this HS assay (S1A Fig). Although *alg-1(gk214)* animals are entirely viable during this time course when maintained at 20˚C (S1B Fig), this strain does exhibit moderate developmental defects [19,24,25], reduced fertility [19,26,27,28] and a shortened lifespan [29]. To avoid pre-existing defects that might make *alg-1(gk214)* more sensitive to HS, we utilized the auxin-inducible degron (AID) system to remove ALG-1 upon HS treatment [30]. Four hours of auxin treatment in the CTL (20˚C) or HS (35˚C) conditions was sufficient to deplete ALG-1 protein, as judged by Western Blotting (Fig 1B). Removal of ALG-1 during HS resulted in reduced levels of survival, comparable to the *alg-1(gk214)* strain (Fig 1A). Given the dramatic auxin-induced depletion of AID::ALG-1, we considered that the reduced viability phenotype could be due to a requirement for ALG-1 during HS or in the recovery period. To test if ALG-1 is needed for recovery from HS, we performed a similar HS viability experiment but instead moved AID::ALG-1 animals to auxin-containing media for the 24 hr recovery period following HS. Loss of ALG-1 after HS resulted in decreased survival compared to WT or the AID::ALG-1 strain in the absence of auxin (Fig 1C). The auxin-induced depletion of AID::ALG-1 was confirmed by Western blot analysis of control and HS-treated animals (Fig 1D). Importantly, auxin treatment (during HS or during recovery) of WT and *alg-1(gk214)* animals did not alter average HS survival of these strains compared to average survival scores for no auxin controls (S1C Fig). Additionally, to control for potential stress induced by moving animals to new plates for auxin treatment, all strains, regardless of condition, were moved to fresh plates. While our results do not rule out a role for ALG-1 activity during HS, they do demonstrate that ALG-1 and, by inference, the microRNA pathway contribute to the recovery phase following HS.

In contrast to the well-studied transcriptional induction of genes that respond to HS, much less is known about how gene expression changes during recovery from this stress [16,33]. There is evidence that in *Drosophila* S2 cells the HS-induced gene Hsp70 is subject to post-transcriptional down-regulation during recovery through a yet to be defined mechanism [15,16,34]. Under our heat shock conditions, *hsp-70 (C12C8.1)* mRNA is up-regulated over 100-fold and recovers to baseline levels within 24 hrs of return to the control temperature (S1D Fig). However, depletion of ALG-1 after HS resulted in significantly higher levels of *hsp-70* mRNA relative to control strains that maintained expression of ALG-1 (Fig 1E). A requirement for ALG-1 in the down-regulation of *hsp-70* following HS raised the possibility that it could be targeted by a miRNA. Using the miRNA target prediction tool TargetScanWorm, we identified two predicted miR-85 binding sites in the 3'UTR of *hsp-70* (Fig 1F) [31]. While additional miRNAs are predicted to target *hsp-70*, miR-85 was the only miRNA with a target site conserved in other nematodes. Furthermore, gene ontology (GO) enrichment analysis using DAVID [32] revealed that the most enriched molecular functions of the 133 predicted miR-85 targets include protein folding chaperone, misfolded protein binding, and heat shock protein binding (Fig 1G). The predicted binding sites within the 3'UTR of *hsp-70* and the potential for targeting other stress response genes focused our attention on miR-85 as a candidate for functioning in the HSR.

## miR-85 regulates genes in stress response pathways

Since miRNA regulation often results in target mRNA degradation [18,35,36], we predicted that *hsp-70* and potentially other stress response target genes might be up-regulated in the absence of miR-85. To test this idea, we performed total RNA sequencing of RNA from last larval stage (L4) WT and *miR-85(n4117)* animals cultured at 20˚C and identified differentially expressed genes using DESeq2 [37]. Consistent with the lack of obvious developmental defects in the *miR-85* deletion mutants [38], few genes were up- (61 genes) or down-regulated (76

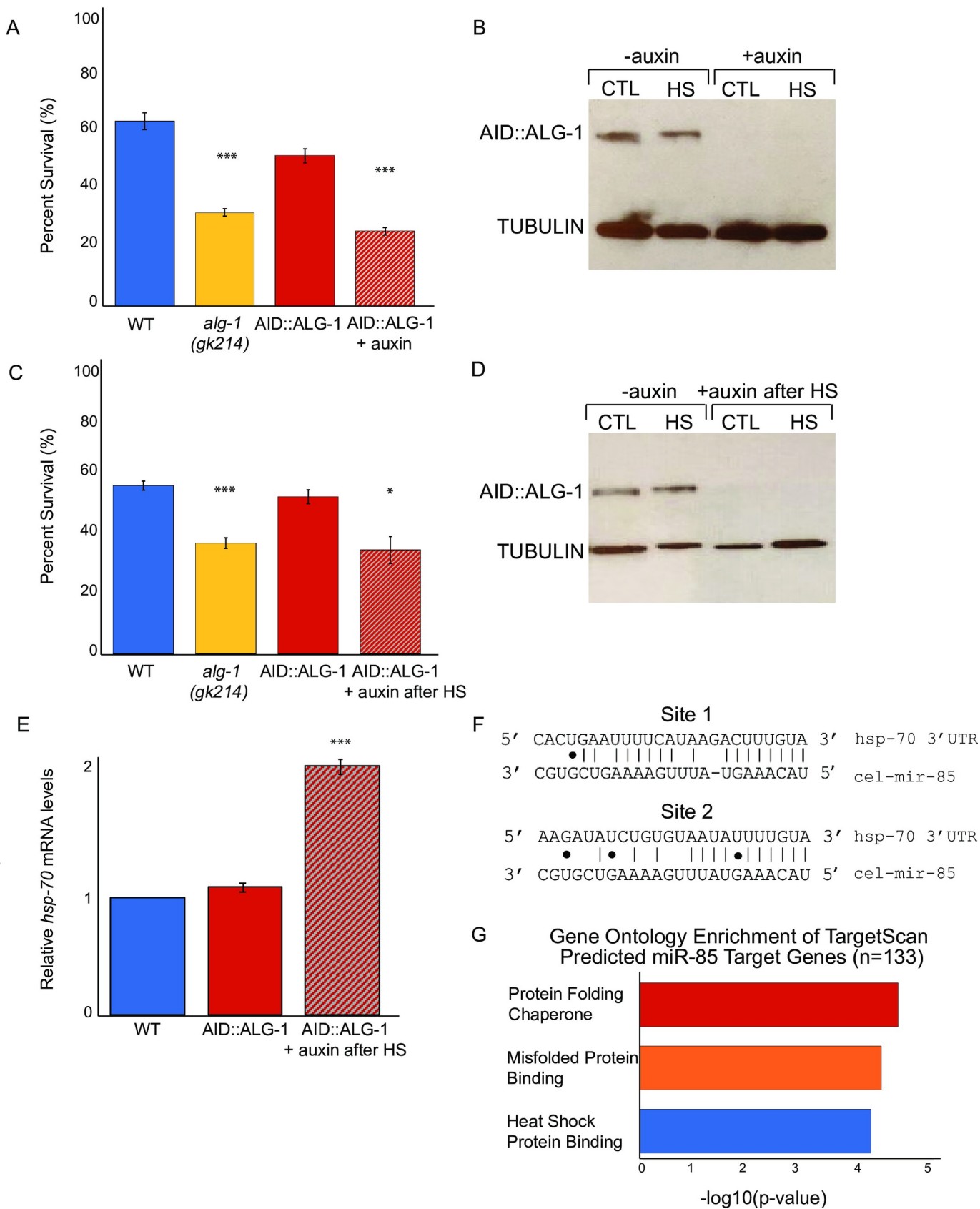

**Fig 1. The microRNA pathway is important for recovery after heat shock.** A. Loss of *alg-1* activity during HS results in reduced viability. Heat shock treatment was performed on synchronized L4 worms for 4 hrs at 35°C and percent survival was determined after 24 hrs recovery at 20°C. The AID::ALG-1 strain was exposed to auxin during the 4 hr HS period. Three biological replicates were performed with at least 50 worms per strain, per condition. Student's t-tests were performed to determine significance compared to WT (\*\*\**P* < 0.001). B. Western blot of ALG-1 protein levels in synchronized L4 AID::ALG-1 worms collected at CTL (20°C) or immediately after 4 hr HS (35°C) in the absence or presence of auxin during the same 4 hr period. C. Loss of *alg-1* activity during recovery after HS results in reduced viability. Heat shock treatment was performed on synchronized L4 worms for 4 hrs at 35°C and percent survival was determined after 24 hrs recovery at 20°C. The AID::ALG-1 strain was placed on auxin immediately after HS during the 24 hr recovery period. Three biological replicates were performed with at least 50 worms per strain, per condition. Student's t-tests were performed to determine significance compared to WT (\**P* < 0.05, \*\*\**P* < 0.001). D. Western blot of ALG-1 protein levels in synchronized AID::ALG-1 worms collected after 24 hrs of recovery post HS treatment or kept at CTL (20°C) in the absence or presence of auxin during the recovery period. E. RT-qPCR of *hsp-70* mRNA after 24 hr recovery from HS in WT and AID::ALG-1 recovered in the absence or presence of auxin. All replicates were normalized to *ama-1* mRNA levels, which are not affected by HS. Three biological replicates were assayed. Student's t-tests were performed to determine significance relative to WT (\*\*\**P* < 0.001). F. Depiction of predicted mir-85 target sites in the *hsp-70* 3'UTR based on Target Scan Worm. Site 1 is conserved and Site 2 is poorly conserved [31]. G. Gene ontology (GO) enrichment analysis of mir-85 predicted targets from Target Scan Worm (n = 133) using DAVID [32].

genes) by at least 2-fold in *miR-85(n4117)* compared to WT (S1 Table). Strikingly, within the set of up-regulated genes, *hsp-70* and several other stress responsive genes were highly over-expressed in the *miR-85(n4117)* mutants (Fig 2A). Additionally, stress response and transit peptide were the only two terms identified among the upregulated genes by GO enrichment analysis using DAVID [32] (Fig 2B). Although the enrichment of these terms was modest, this finding is consistent with a role for miR-85 in regulating stress response genes. Down-regulated genes were enriched for terms associated with collagen, nucleosome assembly and modification, and signaling.

As *hsp-70* was the only predicted miR-85 target up-regulated in *miR-85* Mutants (Fig 2C), it stood out as a prospective candidate for direct regulation by miR-85. To explore this possibility, we used CRISPR-Cas9 to generate a deletion in the *hsp-70* 3'UTR that removes both miR-85 target sites (*hsp-70ΔTS)* (Fig 2D). Like *miR-85* mutants, The *hsp-70ΔTS* strain did not exhibit any obvious developmental abnormalities (S2A Fig). We then examined relative *hsp-70* mRNA levels by RT-qPCR in WT, *miR-85(n4117)*, and *hsp-70ΔTS* L4 stage animals cultured at 20°C. Consistent with the RNA sequencing results, *hsp-70* mRNA was up-regulated ~2-fold in *miR-85(n4117)* compared to WT animals (Fig 2E). The *hsp-70ΔTS* animals exhibited a similar level of *hsp-70* mRNA overexpression relative to WT (Fig 2E). These results show that miR-85 and the 3'UTR of *hsp-70* are important for regulating the levels of *hsp-70* during optimal temperature (20°C) conditions in *C. elegans*.

## miR-85 regulates *hsp-70* during recovery from heat shock

We next asked if miR-85 also plays a role in regulating *hsp-70* as part of the heat shock response. Despite higher basal levels of *hsp-70* mRNA in *miR-85(n4117)* animals (Fig 2D), after 3 hrs of HS at 35°C we detected no significant difference in *hsp-70* mRNA in *miR-85 (n4117)* compared to WT animals (Fig 3A, 0 hr time point). HSP-70 protein also accumulated to similar levels immediately following HS, with no Detectable difference in *miR-85(n4117)* compared to WT (Fig 3B, 0 hr time point). In addition, RNA sequencing after 3 hrs of HS at 35°C revealed very few differences in gene expression with only 4 up- and 9 down-regulated genes (Fig 3C, S1 Table). Given the similar gene expression profiles in WT and *miR-85* strains during HS, we wondered if this congruence might be explained by reduced miR-85 levels in response to HS. This does not seem to be the case as Northern blot analysis showed that miR-85 levels were maintained in HS conditions (Fig 3D). Similar to *miR-85(n4117)*, the *hsp-70ΔTS* animals expressed *hsp-70* mRNA and protein at levels comparable to WT (Fig 3A and 3F, 0 hr time point). These results indicate that the robust induction of *hsp-70* upon HS is not influenced by miR-85 or the *hsp-70* 3'UTR.

**A**

Up and down regulated PCGs in *miR-85(n4117)*

Down PCGs = 76

Up PCGs = 61

- ● P & Log2FC
- ● P-value only
- ● Log2FC only
- ● NS

*hsp-70*

*hsp-16.41*
*hsp-16.2*

-log₁₀p

Log2FoldChange

**B**

Gene Ontology Enrichment of
Up Genes in *miR-85(n4117)* (n=61)

Stress
Response

Transit
Peptide

-log10(p-value)

**C**

miR-85 Predicted
Targets (n=132)

Up PCGs in
*miR-85(n4117)*
(n=60)

*hsp-70*

**D**  *hsp-70 3'UTR:*     Site 1                                   Site 2

5' TTATCTA**CACTGAATTTTCATAAGACTTTGTA**TTTATTTTA**AAGATATCTGTGTAATATTTTGTA**AATAAACATTGGAAAAAATAC 3'

*hsp-70ΔTS*

**E**

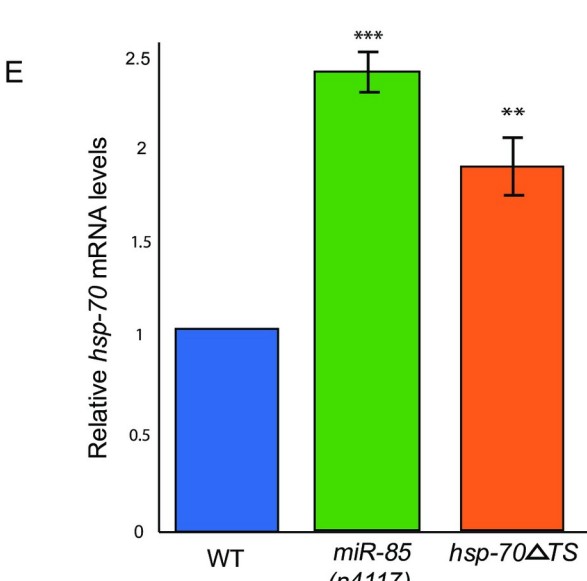

Relative *hsp-70* mRNA levels

*** 

**

WT      *miR-85*      *hsp-70ΔTS*
        *(n4117)*

**Fig 2. mir-85 regulates genes in stress response pathways.** A. Volcano plot of differentially expressed PCGs in WT and *miR-85(n4117)* animals synchronized to L4 at 20°C. Three biological replicates were used for RNA sequencing analysis. Genes were considered significant if they had a base mean > 10, Log2FC +/- 1, and p-adj. < 0.05. B. Gene ontology enrichment analysis of PCGs up-regulated in loss of miR-85 animals vs WT (n = 61) using DAVID [32]. C. Venn-diagram of miR-85 predicted targets from TargetScanWorm [31] and PCGs up-regulated in loss of miR-85 animals. *Hsp-70* is the only gene identified in both groups. D. Depiction of the *hsp-70* 3'UTR with miR-85 target sites in bold and deletion region in the *hsp-70ΔTS* animals marked in red. E. RT-qPCR of *hsp-70* mRNA in WT, *miR-85(n4117)*, and *hsp-70ΔTS* animals. All replicates were normalized to *ama-1* mRNA. Relative expression was calculated by comparing to *hsp-70* expression in WT. Three biological replicates were assayed. Student's t-tests were performed to determine significance compared to WT (**$P < 0.01$, ***$P < 0.001$). Error bars represent SEM.

During HS recovery, however, WT and *miR-85(n4117)* animals showed striking differences in *hsp-70* expression. After return to 20°C, WT animals displayed a rapid decline of *hsp-70* mRNA (Fig 3A) and protein (Fig 3B), following temporal changes previously reported for *hsp-70* during recovery from HS [39]. By 24 hrs after HS, *hsp-70* mRNA had almost returned to non-stress levels in WT but persisted at levels nearly as high as during HS in *miR-85(n4117)* animals (Fig 3A). Furthermore, we found that efficient down-regulation of *hsp-70* during HS recovery was dependent on its 3'UTR. Compared to WT, *hsp-70ΔTS* animals exhibited higher *hsp-70* mRNA and protein levels up to 24 hrs after return to 20°C (Fig 3A and 3F). It should be noted that the antibody used to detect HSP-70 likely reacts with homologs of this protein not expected to be regulated by miR-85 as their 3'UTRs lack binding sites for this miRNA. Thus, the Western blot results are likely an underestimate of the accumulation of HSP-70 in *miR-85 (n4117)* compared to WT animals during recovery from HS. These results demonstrate a requirement for mir-85 and the 3'UTR of *hsp-70* in down-regulating *hsp-70* levels after HS.

## Down-regulation of *hsp-70* by miR-85 is important for recovery from heat shock

While the *miR-85* and *hsp-70ΔTS* strains exhibited no obvious developmental or viability defects (S2A Fig), we wondered if these animals might differ from WT in their ability to survive HS. To examine this possibility, L4 populations of WT, *miR-85(n4117)*, and *hsp-70ΔTS* animals were subjected to 4 hrs of HS at 35°C, and the percent survival was calculated after allowing the animals to recover for 24 hrs at 20°C. Compared to the nearly 60% viability observed in WT, only about 30% of *miR-85(n4117)* and *hsp-70ΔTS* animals survived (Fig 4A). To rule out possible phenotypes associated with *F49E12.8*, which is partially disrupted in *miR-85(n4117)*, we generated a new *miR-85* mutant, *miR-85(ap437)*. This allele contains a short deletion within the pre-miR-85 sequence that does not overlap the *F49E12.8* coding region. The *miR-85(ap437)* animals lacked mature miR-85 (Fig 4B) and phenocopied *miR-85(n4117)*, displaying an average survival around 32% after HS (Fig 4A). Additionally, Mos-1 Mediated Single Copy Insertion (MosSCI) [40] of *miR-85* in the loss-of-function *miR-85(n4117)* mutant (*miR-85(n4117);miR-85+*) restored mature miR-85 expression (Fig 4B) and rescued the HS viability phenotype to WT levels (Fig 4A). Taken together, these findings show that miR-85 and the 3'UTR of *hsp-70* facilitate HS survival.

Given the dramatic mis-regulation of *hsp-70* during recovery from HS in *miR-85(n4117)* and *hsp-70ΔTS* strains (Fig 3A, 3B and 3F), we predicted that the failure to down-regulate *hsp-70* levels after HS might be responsible for the decreased survival of these animals. To test this idea, we used RNA interference (RNAi) to repress *hsp-70* expression during recovery from HS. Compared to vector RNAi control, *hsp-70(RNAi)* resulted in ~10-fold reduction in *hsp-70* mRNA levels in WT, *miR-85(n4117)*, and *hsp-70ΔTS* strains (S3A and S3B Fig). While survival rates for WT animals recovered on vector or *hsp-70(RNAi)* were indistinguishable at about 50%, RNAi depletion of *hsp-70* rescued the reduced viability of *miR-85(n4117)* and *hsp-70ΔTS* strains from ~35% to near WT levels (Fig 4C and 4D). This rescue was not due to a nonspecific RNAi response, as RNAi Against GFP or *dpy-6* did not affect survival of the WT, *miR-85*

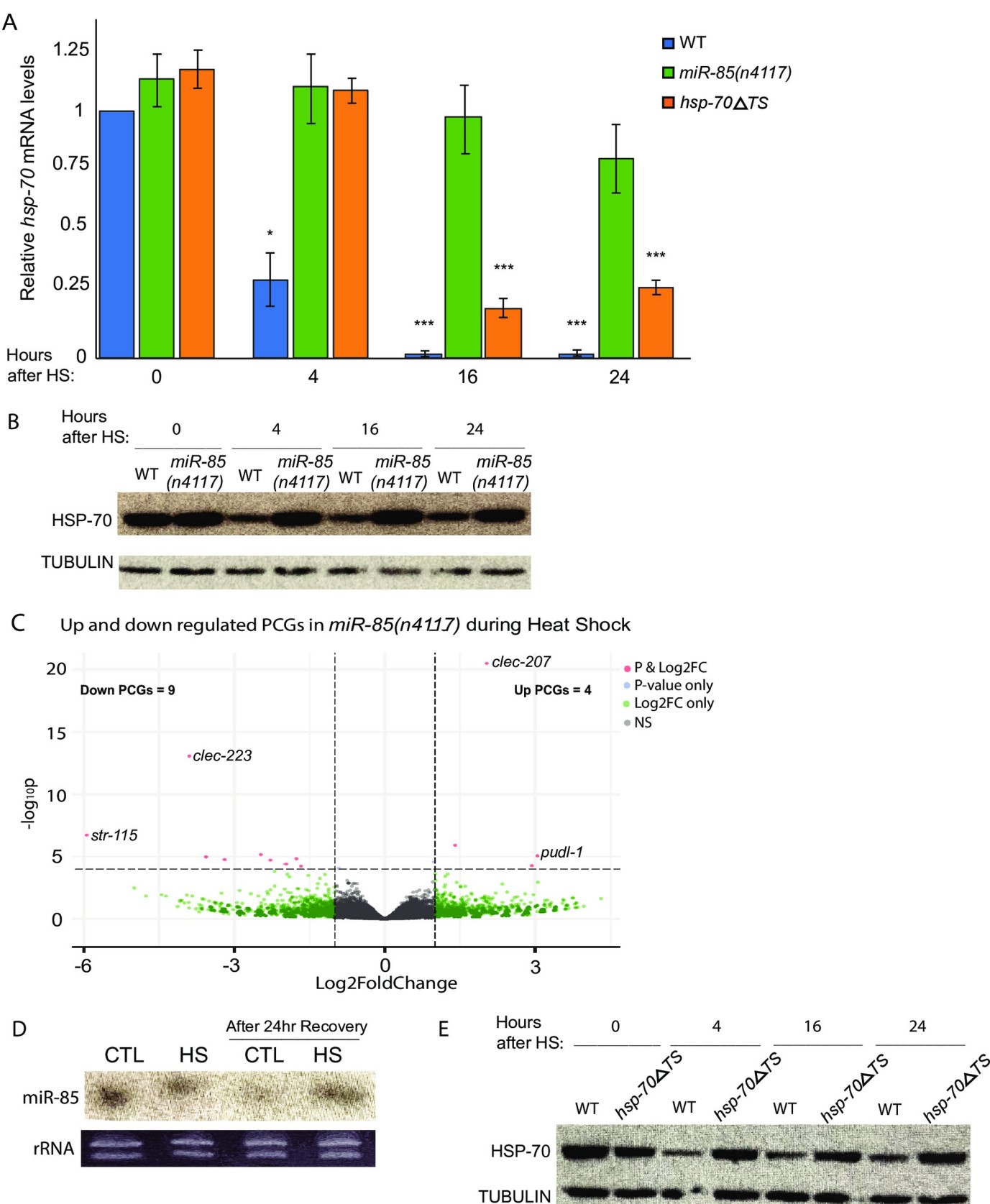

**Fig 3. miR-85 regulates *hsp-70* during recovery from heat shock.** A. RT-qPCR of *hsp-70* mRNA in WT, *mir-85(n4117)*, and *hsp-70ΔTS* animals after 3 hr HS at 35˚C and allowed to recover at 20˚C for 4, 16, and 24 hrs at 20˚C. All replicates were normalized to *ama-1* mRNA. Three biological replicates were assayed. Student's t-tests were performed to determine significance relative to levels immediately after HS (0 hr) in WT animals (*$P < 0.05$, ***$P < 0.001$). Compared to the WT 0 hr time point, *mir-85(n4117)* showed no significant difference in *hsp-70* mRNA levels through the recovery time course. Error bars represent SEM. B. Western blot of HSP-70 protein levels in WT and *miR-85(n4117)* animals subjected to 3 hr HS at 35˚C and allowed to recover at 20˚C for 4, 16, and 24 hrs at 20˚C. Tubulin was used as a loading control. C. Volcano plot of differentially expressed PCGs in WT and *miR-85(n4117)* animals synchronized to L4 and heat shocked for 3 hrs at 35˚C. Three biological replicates were used for RNA sequencing analysis. Genes were considered significant if they had a base mean > 10, Log2FC +/- 1, and p-adj. < 0.05. D. Northern blot of RNA from L4 synchronized WT animals maintained at 20˚C (CTL) or after 3 hrs of HS at 35˚C, and collected 24 hrs later from CTL and HS animals recovered at 20˚C to detect miR-85-3p; ethidium bromide staining of rRNAs shows similar levels of loaded RNA for each sample. E. Western blot of HSP-70 protein levels in WT and *hsp-70ΔTS* animals subjected to 3 hr HS at 35˚C and allowed to recover at 20˚C for 4, 16, and 24 hrs at 20˚C. Tubulin was used as a loading control.

*(n4117)* or *hsp-70ΔTS* strains after HS. These results indicate that overexpression of *hsp-70* during recovery from HS contributes to the diminished survival of animals lacking miR-85 or the miR-85 target sites in the *hsp-70* 3'UTR. Furthermore, they highlight the importance of down-regulating *hsp-70* levels after HS and reveal a new role for the miRNA pathway in this process (Fig 4E).

## Discussion

Here we demonstrate a novel role for the microRNA pathway in heat shock recovery in *Caenorhabditis elegans*. We identified a specific miRNA, miR-85, that represses the expression of stress response genes, including a HS-induced chaperone, *hsp-70*. We show that animals lacking miR-85 or miR-85 target sites within the 3'UTR of *hsp-70* overexpress *hsp-70* under optimal temperature (20˚C) conditions. Upon HS, though, this difference vanishes as the expression of *hsp-70* and other HS-induced genes is robustly activated at the transcriptional level. However, during recovery from HS, the normally rapid decline in *hsp-70* expression falters in the absence of miR-85 or its target sites in the *hsp-70* 3'UTR. Moreover, we present evidence that this over-expression of *hsp-70* during HS recovery reduces organismal viability. Our study reveals the importance of downregulating *hsp-70* levels after HS in an intact animal and highlights the role of miR-85 in facilitating this process.

### Role of the miRNA pathway in stress responses

Across animal species, disruption of a single miRNA rarely results in overt phenotypes [18,41]. However, the majority of studies have been performed under well-controlled laboratory conditions that seldom reflect stressful events encountered in the wild. When animals are challenged by bouts of stress, specific miRNAs have emerged as important regulatory tools that add robustness to developmental pathways and promote survival programs [41,42]. One striking example is illustrated by the role of miR-7 in *Drosophila*. Flies lacking this highly conserved miRNA exhibit no apparent abnormalities under controlled laboratory conditions [43]. However, when miR-7 mutants were subjected to a series of temperature fluctuations eye development failed [44]. Examples can also be found in tumor microenvironments, where localized cellular stress can reveal roles for specific miRNAs [45]. In mice, deletion of miR-10a caused no obvious defects until the mutant strain was challenged with a model of intestinal neoplasia [46]. In this background, the lack of miR-10a was associated with a dramatic increase in the development of adenomas in female mice [46]. There is also extensive evidence of miRNAs acting as key regulators in plants responding to various forms of stress, again unveiling previously unappreciated functions that are not evident under optimal growth conditions [47].

In *C. elegans*, few miRNA-associated phenotypes have been identified under normal laboratory conditions [38]. However, stress-responsive miRNAs have been reported [20–22] and a few miRNAs, such as miR-71 and miR-246, have been shown to influence viability after a bout

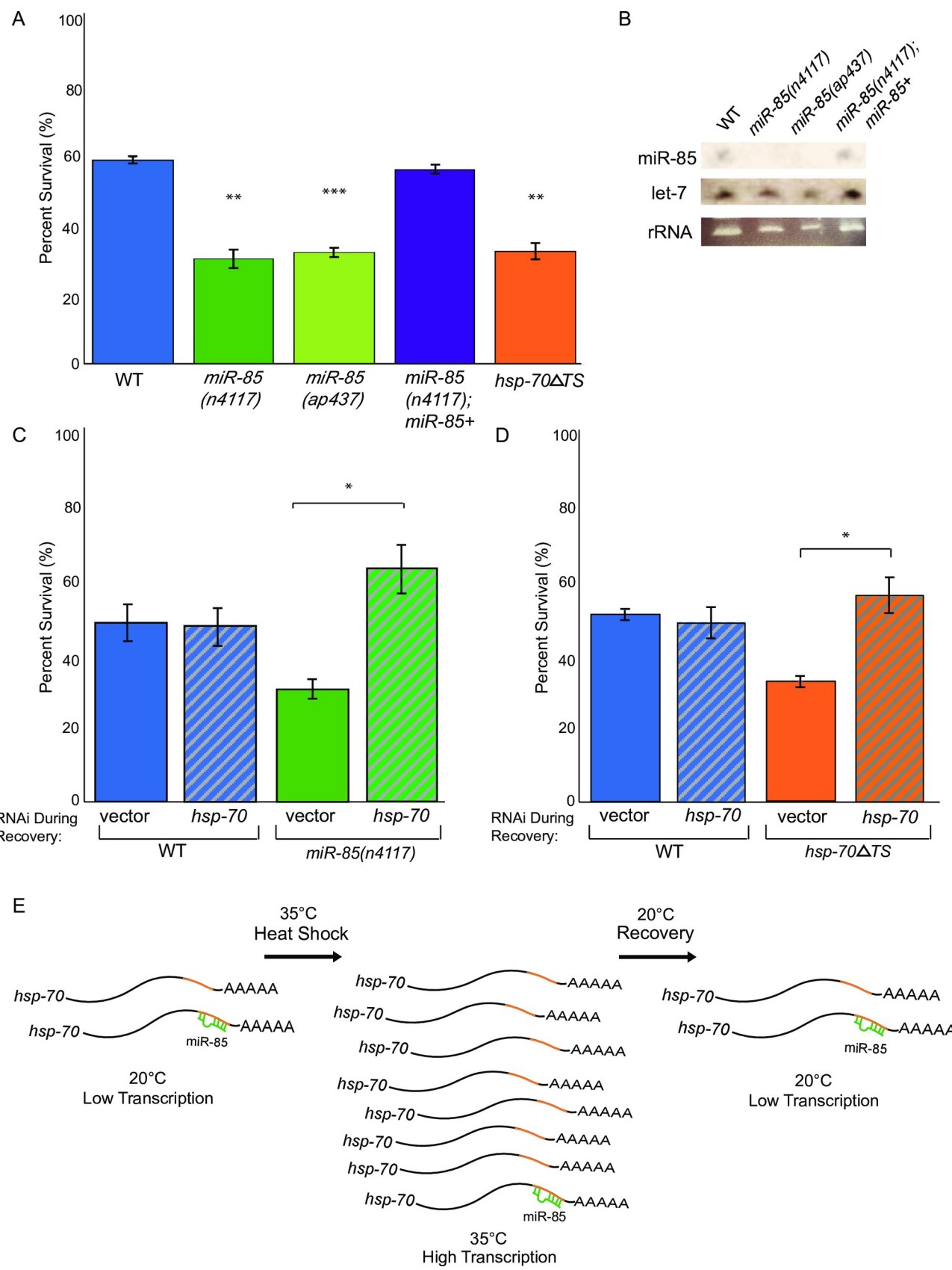

**Fig 4. Down-regulation of *hsp-70* by miR-85 is important for recovery from heat shock.** A. Heat shock viability of WT, *miR-85(n4117)*, *miR-85 (ap437)*, *miR-85(n4117); miR-85+*, and *hsp-70ΔTS* animals subjected to 4 hr of HS at 35˚C. Percent survival was determined after 24 hrs recovery at 20˚C. *miR-85(ap437)* was generated using CRISPR-Cas9 to disrupt the production of mature miR-85. The *miR-85(n4117)*; *miR-85+* rescue strain was generated using MosSci. The rescue strain genotype is *knuSi832[pNU2156(miR-85 in cxTi10882, unc-119(+))] IV; unc-119(ed3) III; miR-85(n4117)*. Three blinded biological replicates were performed with at least 50 worms per strain, per condition. Student's t-tests were performed to determine significance compared to WT (\*\**P* < 0.01, \*\*\**P* < 0.001). Error bars represent SEM. B. Northern blot of L4 synchronized WT, *miR-85(n4117)*, *miR-85 (ap437)* and *miR-85(n4117); miR-85+* strains probed for miR-85-3p and let-7 as a control to ensure strains were developmentally synchronized to the L4 stage; ethidium bromide staining of rRNA shows similar levels of loaded RNA for each sample. C. Following 4 hr of HS at 35˚C, WT and *miR-85(n4117)* animals were subjected to vector *RNAi* or *hsp-70 RNAi* and percent survival was determined after 24 hrs recovery at 20˚C. Three blinded biological replicates were performed with at least 50 worms per strain, per condition. Student's t-tests were performed to determine significance for each strain relative to survival on *vector RNAi* (\**P* < 0.05). D. Following 4 hr of HS at 35˚C, WT and *hsp-70ΔTS* animals were subjected to vector *RNAi* or *hsp-70 RNAi* and percent survival was determined after 24 hrs recovery at 20˚C. Three blinded biological replicates were performed with at least 50 worms per strain, per condition. Student's t-tests were performed to determine significance for each strain relative to survival on *vector RNAi* (\**P* < 0.05). E. Model for the role of miR-85 role in regulating *hsp-70*. At 20˚C, miR-85 represses *hsp-70* by binding to complementary sites in the 3'UTR of its mRNA. Heat shock induces transcriptional up-regulation of *hsp-70* and the very high levels of *hsp-70* mRNA override the regulatory potential of miR-85. Upon return to 20˚C, the transcriptional induction ceases and miR-85 aids in the rapid down-regulation of the abundant *hsp-70* mRNAs.

of stress [23]. Nonetheless, there remains much to learn about the precise roles and targets of miRNAs in *C. elegans*, underscoring the importance of investigating miRNA function under conditions that mimic challenges faced in the natural world.

Here, we demonstrate a role for the miRNA pathway in regulating heat shock survival in *C. elegans*. The auxin-inducible degron system allowed us to deplete ALG-1 during the HS period or immediately following it. This method enabled us to conclude that ALG-1 functions in the HSR, but it is unclear if this role is restricted to the recovery phase or also engaged during high temperatures. The ability of the miRNA complex to regulate gene expression during HS is an open question. In mammalian cell culture, stress can result in modification of Argonaute proteins with poly(ADP-ribose) moieties and accumulation in Stress Granules (SGs), which store translationally inert mRNAs [48,49]. These events are associated with relief of miRNA-mediated repression [49]. There is also evidence that arsenite stress induces changes in target interactions by Argonaute that, instead, might enhance translational repression [50]. A further consideration is the general stabilization and translational repression of pre-existing mRNAs upon heat shock, which is a conserved response across organisms including those that lack the miRNA pathway, such as *S. cerevisiae* and *E. coli* [51–53]. Thus, the importance of miRNA-mediated gene regulation during elevated temperatures remains an open question.

## Post-transcriptional regulation of *hsp-70*

From bacteria to humans, dramatic up-regulation of *hsp-70* is a conserved feature of the heat shock response [54]. While many organisms also have constitutively expressed *hsp-70* homologs, the stress induced members are distinguished by their rapid induction in response to increased temperature [55]. The rapid accumulation of *hsp-70* triggered by HS and other stresses is regulated at the transcriptional level by Heat Shock Factor 1 (HSF1) [9]. Upon return to non-stress conditions, this transcriptional program ceases and, at least in some organisms, a post-transcriptional mechanism acts to reduce *hsp-70* to near basal levels [16]. Studies in human and *Drosophila* cells have shown that under control temperatures, *Hsp70* mRNA is relatively unstable [56,57]. During HS though, accelerated decay of *Hsp70* mRNA appears to be suspended, allowing high levels of the transcripts to accumulate for translation. Then, during recovery rapid degradation of *Hsp70* mRNA resumes [56,57]. It has been shown in *Drosophila* cells that degradation of *Hsp70* mRNA is achieved through efficient deadenylation and subsequent 5'-decapping and decay, and, consistent with previous studies, this process appears to arrest during stress and resume during recovery [17,58]. Regulation of *Drosophila Hsp70* mRNA stability is mediated by its 3'UTR [16,17,34]. Inspection of the *hsp-*

*70* 3'UTR sequence revealed well-conserved "instability motifs" (AUUUA), which were hypothesized to contribute to the targeted deadenylation of *Hsp70* [16]. However, disruption of these elements in reporters fused to the *Hsp70* 3'UTR did not affect mRNA decay rates in *Drosophila* cells, leaving the precise destabilizing elements within the 3'UTR yet to be defined [17].

Our study establishes a regulatory role for the 3'UTR of *hsp-70* in *C. elegans*. Our results are consistent with a model whereby miR-85 binds target sites in the *hsp-70* 3'UTR to promote mRNA degradation (Fig 4E). Despite multiple attempts, we failed to generate precise mutations that only changed the miR-85 target sites in the 3'UTR of *hsp-70*, so it remains possible that additional 3'UTR regulatory elements are disrupted in *hsp-70ΔTS* animals. While *hsp-70* mRNA levels were elevated to similar extents in *miR-85* or *hsp-70ΔTS* mutants under control temperatures and up to 4 hours into HS recovery, levels were markedly higher at the later recovery time points in animals lacking miR-85 (Fig 3A, 16, 24 hr time points). This difference may stem from mis-regulation of other miR-85 targets in the *miR-85* mutant, which could indirectly affect *hsp-70* expression during recovery.

The apparent pause in miRNA-mediated regulation of *hsp-70* during HS in *C. elegans* is reminiscent of the halt in *Hsp70* mRNA decay observed in *Drosophila* and human cells [56,57]. Given the parallels, it is tempting to speculate that the miRNA pathway may also contribute to the post-transcriptional regulation of *Hsp70* in other organisms. As described above, modification and re-localization of Argonaute during stress could limit the repressive ability of the miRNA complex and potentially release *hsp-70* mRNA for maximal expression. Another mechanism that can override effective regulation by the miRNA pathway is an over-production of target transcripts [41]. As miR-85 levels remain constant during HS, it is possible that the bolus of *hsp-70* mRNA and possibly other miR-85 targets induced by transcription exceeds a threshold where miRNA regulation contributes little to overall target levels.

### Down-regulation of *hsp-70* after heat shock is important for survival

We found that maintenance of high *hsp-70* levels after HS negatively impacts survival in *C. elegans*. Since depletion of *hsp-70* mRNA by RNAi could compensate for the lack of miR-85 mediated regulation, overexpression of *hsp-70* is likely the primary cause of reduced viability in *miR-85* or *hsp-70ΔTS* mutants subjected to HS. HSP-70 is a molecular chaperone that can promote protein folding, refolding, disaggregation or degradation, activities under high demand in elevated temperatures [12]. Given these protective roles, why then would excess HSP-70 be detrimental? This is a longstanding question as previous studies have shown that constitutive overexpression of *Hsp70* in *Drosophila* cells reduced growth rates [59] and in mice negatively impacted development and lifespan [60]. Given its many roles in the cell under normal growth conditions, which include facilitation of protein folding, translocation across membranes, complex assembly and disassembly, and regulation of protein activity and stability [12], there are several possibilities that could explain the danger of excess Hsp70. The ability of Hsp70 to interact promiscuously with a variety of clients could result in a general disruption in protein homeostasis where deviations in folding or complex formation dynamics lead to aberrant or unstable complexes. High levels of *Hsp70* could also change the stoichiometry with binding partners, such as the transcription factor HSF1 or the endoribonuclease Ire that mediates ER stress, reducing their active concentrations [60,61]. There is also evidence that Hsp70 interacts with lipids in mammalian culture cells, and it has been speculated that aberrant membrane interactions could contribute to the cellular toxicity caused by excess Hsp70 [61,62]. While it is yet to be determined how high levels of HSP-70 hinder the ability of *C. elegans* to

recover from HS, this finding highlights the importance of down-regulating *hsp-70* expression post stress.

Curiously, the overexpression of *hsp-70* observed in *miR-85* or *hsp-70ΔTS* mutants was not associated with developmental abnormalities in animals cultured at ambient temperatures. Despite the up-regulation of several stress-related genes in the *miR-85* mutants, the development of these animals is indistinguishable from wildtype. These observations suggest that under ideal conditions the robustness of developmental programs is not dependent on regulation by miR-85. However, this miRNA was found to be required for recovery from HS. Thus, environmental perturbations likely to be encountered in the wild, such as elevated temperatures, can reveal roles for miRNA genes previously considered to be dispensable. Furthermore, our study demonstrates that repression of *hsp-70* by miR-85 promotes survival after HS in *C. elegans*, establishing an important role for the miRNA pathway in post stress gene regulation.

## Materials and methods

### Nematode culture and heat shock viability assays

*C. elegans* strains were cultured under standard conditions and synchronized by hypochlorite treatment [63]. Heat shock viability assays were performed by plating synchronized L1 worms on NGM plates seeded with OP50 that were grown for 44 hrs at 20°C before raising the temperature to 35°C for 4 hrs of heat shock. Worms were recovered for 24hrs at 20°C before scoring. When possible, assays were blinded before synchronization and were unblinded only after scoring viability. *Alg-1(gk214)* worms were grown for 47 hrs before HS treatment as they are slightly developmentally delayed [64]. The auxin-induced degradation was performed by moving worms to NGM plates supplemented with 1% auxin (indole-3-acetic acid) during either 4 hrs of HS or 24 hrs of recovery. Numerical data are provided in S2 Table.

### Western blotting

Western blotting was performed as previously described using mouse monoclonal antibodies against Tubulin or FLAG (Sigma), and rabbit polyclonal antibodies against Hsp70 (Proteintech) [65,66].

### RT-qPCR

Quantitative RT-PCR analyses of mRNA (SYBR Green) levels were performed according to manufacturer's instructions using the QuantSudio machine (ABI Biosystems). The housekeeping gene, *ama-1*, was used for normalization of experimental Ct values. Three biological replicates were performed with three technical replicates for each target gene. Numerical data are provided in S2 Table.

Primer sequences:
*ama-1* Forward: 5'–CACTGGAGTTGTCCCAATTCTTG– 3'
*ama-1* Reverse: 5'–TGGAACTCTGGAGTCACACC– 3'
*hsp-70* Forward: 5'–CCGCTCGTAATCCGGAGAATACAG– 3'
*hsp-70* Reverse: 5'–CAACCTCAACAACGGGCTTTCC– 3'

### Northern blotting

PAGE northern blotting was performed as previously described using IDT StarFire probes for cel-miR-85-3p and let-7 [66]. let-7: AACTATACAACCTACTACCTCA/3StarFire /miR-85-3p: GCACGACTTTTCAAATACTTTGTA/3StarFire/

## RNA sequencing

N2 wildtype (WT) and *miR-85(n4117)* worms were grown to L4 stage (47 hrs at 20˚C) under standard growth conditions [63].The experimental group was subjected to heat stress by raising the temperature to 35˚C for 3 hrs after developing for 44 hrs at 20˚C. Animals from the control (20˚C) and experimental group (35˚C) were collected, snap-frozen and total RNA was isolated using a standard TRIzol (Life Technologies) RNA extraction. cDNA sequencing libraries from three independent biological replicates were prepared from 1ug of total RNA (with RIN > 8.4) using the standard protocol from the Illumina Stranded TruSeq RNA library prep kit. Ribosomal RNA was removed prior to library preparation using RiboZero Gold (Illumina). cDNA libraries were sequenced on the Illumina High Seq 4000 (SR75). Libraries were at least 28 million reads per sample. Reads were aligned to the *C. elegans* genome WS235 using STAR and the average percent of uniquely mapped reads was 84.4% [67]. Aligned reads were sorted using Samtools [68]. Reads were counted using FeatureCounts and Ensembl 88 gene annotations [69]. Differential gene expression was determined using DESeq2 with default parameters [37]. Classes of genes were filtered, and protein-coding genes (PCGs) were used for downstream analysis. PCGs with +/- 1 Log2FC, a basemean of at least 10, and adjusted p-value of < 0.05 were considered significantly mis-regulated (S1 Table). Volcano plots were generated using EnhancedVolcano [70]. Gene ontology enrichment was determined using DAVID Gene Ontology Analysis of the PCGs determined to be significantly up or down-regulated in *miR-84(n4117)* compared to WT animals by RNA sequencing [32].

## RNAi

Feeding RNAi was performed as previously described [71]. Briefly, animals were moved to either empty vector, *GFP*, *dpy-6* or *hsp-70* RNAi plates during a 24 hr recovery period at 20˚C after a 3 hr HS for RT-qPCR analysis or after 4 hr HS for viability experiments. RNAi plates were supplemented with 1mM of IPTG, tetracycline (12.5ug/ml), and ampicillin (100ug/ml). RNAi bacteria was grown for 16 hrs at 37˚C and concentrated 10x before adding to plates. Numerical data are provided in S2 Table.

## Strains

The following strains were used in this study: wild type (WT) N2 Bristol, QK155 (AID::ALG-1): *alg-1(xk20 | 4xflag::degron::alg-1) X; ieSi57 II* (a gift from the John Kim Lab), VC446 *alg-1 (gk214)*, RB574 *alg-2(ok304)*, PQ629 *miR-85(n4117)*. PQ610 *miR-85(ap437)* and PQ659 *hsp-70ΔTS(ap443)* strains were generated using CRISPR/Cas9 with a single guide RNA and backcrossed to N2 three times. The miR-85 rescue strain PQ602 was made by crossing *miR-85 (n4117)* to COP2068: knuSi832[pNU2156 (miR-85 in cxTi10882, unc-119(+))] IV; unc-119 (ed3) III), using a MosSCi integration system (Nemametrix) [40].

## Supporting information

**S1 Table. List of up- and down-regulated PCGs from RNA sequencing experiments at control (20˚C) and heat shock (35˚C) conditions in WT versus *miR-85(n4117)* animals.** Three independent biological replicates were used for RNA sequencing.
(XLSX)

**S2 Table. Numerical data underlying graphs and summary statistics.**
(XLSX)

**S1 Fig. Viability assay controls and *hsp-70* RT-qPCR during heat shock and recovery.** A. Percent survival of WT, *alg-1(gk214)*, and AID::ALG-1 animals grown at 20˚C for 72 hrs. Three biological replicates were performed with at least 50 worms per strain. Error bars represent SEM. B. Percent survival of synchronized WT and *alg-2(ok304)* animals grown at 20˚C for 44 hrs before heat shock for 4 hrs at 35˚C. Viability was scored after 24 hrs of recovery at 20˚C. Three biological replicates were performed with at least 50 worms per strain. Error bars represent SEM. C. Percent survival of synchronized WT, *alg-1(gk214)*, and AID::ALG-1 animals grown to stage L4 before being moved to auxin-containing media either for 24 hrs at 20˚C, during 4 hrs HS, or during 24 hrs recovery after HS. Three biological replicates were performed with at least 50 worms per strain. Error bars represent SEM. D. RT-qPCR of *hsp-70* mRNA levels in WT animals subjected to 3 hr HS and allowed to recover for 24hrs. All replicates were normalized to *ama-1*. Three biological replicates were assayed. Student's t-tests were performed to determine significance relative to *hsp-70* mRNA expression compared to HS (***$P < 0.001$). Error bars represent SEM.
(PDF)

**S2 Fig. Viability assay at control temperatures (20˚C).** A. Percent survival of WT, *miR-85 (n4117)*, *miR-85(ap437)*, and *hsp-70ΔTS* animals grown at 20˚C for 72 hrs. Three blinded biological replicates were performed with at least 50 worms per strain. Error bars represent SEM.
(PDF)

**S3 Fig. RT-qPCR of RNAi knock down and non-specific RNAi heat shock viability assay controls.** A. RT-qPCR of *hsp-70* mRNA levels in WT and *miR-85(n4117)* animals subjected to HS and allowed to recover on *vector RNAi* or *hsp-70 RNAi* for 24 hrs. All replicates were normalized to *ama-1* mRNA. Three biological replicates were assayed. Student's t-tests were performed to determine significance relative to *hsp-70* mRNA expression immediately after HS for each strain (***$P < 0.001$). Error bars represent SEM. B. RT-qPCR of *hsp-70* mRNA levels in WT and *hsp-70ΔTS* animals subjected to HS and allowed to recover on *vector RNAi* or *hsp-70 RNAi* for 24 hrs. All replicates were normalized to *ama-1* mRNA. Three biological replicates were assayed. Student's t-tests were performed to determine significance relative to *hsp-70* mRNA expression immediately after HS for each strain (*$P < 0.05$, **$P <0.01$). Error bars represent SEM. C. Following 4 hr of HS at 35˚C, WT, *miR-85(n4117)*, and *hsp-70ΔTS* animals were subjected to *GFP* and *dpy-6 RNAi* and percent survival was determined after 24 hrs recovery at 20˚C. Three blinded biological replicates were performed with at least 50 worms per strain, per condition. Student's t-tests were performed to determine significance for each strain relative to survival on *vector RNAi* (*$P < 0.05$, **$P < 0.01$, ***$P < 0.001$). Error bars represent SEM.
(PDF)

## Acknowledgments

We thank Dr. Cindy Voisine and members of the Pasquinelli Lab for helpful discussions and critical reading of the manuscript. Sequencing of libraries used for RNA-seq analyses was conducted at the IGM Genomics Center, University of California, San Diego, CA. Some strains were provided by the Caenorhabditis Genetics Center (CGC).

## Author Contributions

**Conceptualization:** Delaney C. Pagliuso, Amy E. Pasquinelli.

**Data curation:** Delaney C. Pagliuso, Devavrat M. Bodas.

**Formal analysis:** Delaney C. Pagliuso.

**Funding acquisition:** Amy E. Pasquinelli.

**Investigation:** Delaney C. Pagliuso, Devavrat M. Bodas, Amy E. Pasquinelli.

**Methodology:** Delaney C. Pagliuso, Amy E. Pasquinelli.

**Project administration:** Amy E. Pasquinelli.

**Supervision:** Amy E. Pasquinelli.

**Validation:** Delaney C. Pagliuso.

**Visualization:** Delaney C. Pagliuso, Amy E. Pasquinelli.

**Writing – original draft:** Delaney C. Pagliuso.

**Writing – review & editing:** Delaney C. Pagliuso, Devavrat M. Bodas, Amy E. Pasquinelli.

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
