## [Decision Letter · Decision Letter 0]

24 May 2021

Dear Dr Pasquinelli,

Thank you very much for submitting your Research Article entitled 'Recovery from Heat Shock Requires the MicroRNA Pathway in Caenorhabditis elegans' to PLOS Genetics.

The manuscript was fully evaluated at the editorial level and by independent peer reviewers. The reviewers appreciated the attention to an important topic but identified some concerns that we ask you address in a revised manuscript. We therefore ask you to modify the manuscript according to the review recommendations. Your revisions should address the specific points made by each reviewer.

[LINK]

Yours sincerely,

Eric A Miska, PhD

Associate Editor

PLOS Genetics

Gregory P. Copenhaver

Editor-in-Chief

PLOS Genetics

Reviewer's Responses to Questions

**Comments to the Authors:**

Reviewer #1: Pagliusio and collaborators report here the involvement of the microRNA pathway in the recovery of the nematode C. elegans upon heat shock. More particularly, the authors show that the Argonaute ALG-1, and more particularly a specific microRNA, miR-85, is needed for this biological process. Genetic and molecular experiments also suggest that miR-85 is involved in the regulation of hsp-70, a gene that needs to be down-regulated for allowing recovery and survival of the animal after heat shock.

Overall, this is a very interesting study that shed light on a new gene regulatory pathway important for animal recovery upon heat shock that involved a specific microRNA. While the experiments presented here strongly support the contribution of ALG-1 and miR-85 in this process, it is still unclear how the microRNA pathway precisely contributes to heat shock recovery as the miR-85/hsp-70 regulatory axis appears not sufficient to fully explain this biological process. Therefore, it is important to perform the following experiments to help better define how miR-85 controls this stress response.

-In silico identification of putative miR-85 targets and the comparative RNA sequencing analysis infer the potential regulation by miR-85 of other key mRNAs, besides hsp-70, involved in stress response. Furthermore, the data presented in Figure 3 suggest that miR-85 is needed for the acute clearance of hsp-70 mRNA, but at a later time point, the down-regulation of hsp-70 does not seem to directly involve miR-85 as hsp-70 mRNA deleted for miR-85 binding sites decreased substantially 16hr after heat shock. As mir-85 mutant retains high hsp-70 level after heat shock, other mRNAs targeted by miR-85 likely contributed to its down-regulation. Therefore, it will be important to test the involvement of other potential miR-85 targets in this process as performed for hsp-70. A subset of genes to be tested can be easily identified from the ones found by comparative RNA sequencing.

-To better compare the effects on hsp-70 mRNA level detected in mir-85 mutant and in animals expressing hsp-70 without miR-85 target sites during heat shock recovery, plotting the data on the same graph will be necessary. That will allow the readers to appreciate the magnitude of effects seen in both conditions and determine the importance of the direct miR-85 regulation of hsp-70.

-To better define the contribution of miR-85 during heat shock recovery, it will be warranted to monitor its level at the same time points where hsp-70 mRNA level has been surveyed (0, 4, 16 and 24h).

Besides those crucial issues, the following also need to be corrected:

-P.16, first paragraph: Figure 3E and 3A should be referred to rather than Figure 4E and 4A

-Figure 1F: Please show the complete base-pairing of the miR-85 with the site 2 of hsp-70 3’UTR

-P.6, line 112: Besides the paper listed, the fertility of alg-1(gk214) animals has directly been surveyed in Bukhari et al., Cell Research 2012 paper. Please add this citation.

Reviewer #2: The manuscript titled “Recovery from Heat Shock Requires the MicroRNA Pathway in

Caenorhabditis elegans” by Pagliuso et al. shows that miRNA pathway and specifically miR-85 in C. elegans is required to regulate the expression of the conserved heat shock protein HSP-70 during recovery from heat shock. This is a very interesting manuscript, well written and well presented. One particularly important aspect of the manuscript is that the authors focus on the recovery from heat shock. While there is many research publications describing stress response, attention to recovery from stress and how this part of the physiology is regulated has been lacking. In addition, the manuscript identifies an important biological role for the miRNA pathway in recovery from stress. The authors provide multiple lines of evidence of miRNA regulation of HSP-70 during the recovery phase by using miRNA miR-85 deletions lines, removal of sequences targeted by miR-85 within the HSP-70 3ʹ UTR.

I recommend the publication of the manuscript at PLoS Genetics once the following issues are addressed either through additional data analysis or further clarification within the manuscript text.

1- It is not sufficiently clear why the authors focused on hsp-70 among all the other heat shock proteins to begin with. Have the authors investigated the response of other HS factors?

2- Authors use TargetScanWorm to identify two predicted miR-85 binding sites. Did the authors identify other miRNA binding sites in hsp-70 UTR?

3- Regarding the RNA-Seq experiment of miR-85 mutants, I couldn’t see a link to the deposited data in materials & methods. The authors should also include some QC data on their RNA-Seq, RNA RIN values if available, read counts of libraries, % mapped, %uniquely mapped, % multi-mapper, PCA analysis between samples and to RIN values. If any custom codes has been used these should be provided as well to enable independent validation of the data.

4- Line 162, Do the authors see enrichment of predicted miR-85 targets among the up regulated genes in comparison to the down-regulated genes?

5- Line 164, Can the authors compare the GO term enrichment analysis between the up and down regulated genes to see if the enrichment in up-regulated genes is specific to this set? How these Go enrichments are calculated is not clarified in materials and methods.

6- Page 7, line 139 - Authors mention that upon HS hsp-70 mRNA is up-regulated over 100 fold. However, in Figures 1 & 2, hsp-70 mRNA levels are only 2 fold higher when miR-85 regulation doesn’t happen during recovery. Can the authors clarify these differences and discuss more clearly how this is sufficient for the protein expression changes and the phenotypes?

Reviewer #3: Summary:

In their manuscript "Recovery from Heat Shock Requires the MicroRNA Pathway in Caenorhabditis elegans", Pasquinelli and colleagues analyze the requirement of miRNAs in a robust recovery from heat shock. They report that alg-1 mutants have reduced survival and retain elevated Hsp70, and that direct regulation of hsp-70 by miR-85 is required during recovery from heat stress.

This is a short and focused study of a miRNA-target pair that has physiologically relevant effects, and exploits the clean genetic analysis that is now possible in the worm. The work is generally well-done and uses excellent in vivo reagents (AID-Alg1, miRNA mutants and 3'UTR site mutants). It follows up on a line of thinking in the miRNA field in general about miRNA control in stress response, but these genetic tests are superior to most prior works and so are a good fit for Plos Genetics. I am supportive of publication following addressing a few issues raised.

Key findings:

• alg-1 mutants, which are known to have are various defects, are sensitive to heat shock. They further use temporally controled AID system to show that Alg1 is required concomitant with heat shock, avoiding pre-existing defects.

• hsp70 mRNA is induced 100X by heat shock, and returns to baseline by 24hrs. It's proposed not only due to transcriptional regulation, but post-transcriptional, since hsp70 is up about 2 fold in AID::ALG-1. miR-85 is a direct regulator of hsp70, since ∆mir85 mutants and an hsp70∆miR-85 sites (∆TS) strain shown 2-fold elevation of hsp70. While miR-85 and and hsp70∆TS strains seem normal, they have defective recovery from HS. The ∆mir85 can be rescued by miR-85 transgene, and hsp70-RNAi can increase survival of mir-85 and hsp70∆TS after heat shock.

Major concerns:

• I am not sure how precisely the hsp70-RNAi can result in depletion of Hsp70 protein. The rescue of mir-85 and hsp70∆TS mutants by hsp70-RNAi is a nice demonstration of the genetic impact of elevated Hsp70 in these mutants. However, its surprising since Hsp70 is a central coordinator of heatshock response. I think it could be quite difficult to precisely get back to a normal Hsp70 level by RNAi in these mutants and thus I think HS response could be negatively impacted. I may miss it, but I didn't see Hsp70 timecourse Western in these conditions (Figure 4) to know its induction and downregulation in hsp70-RNAi+mutants.

• This is a very short report, which is ok. However, as they mention, organisms will face many stressors during life ("oxidative stress, pathogen infection, starvation, heat stress" etc. Let us add COVID19 life...) I think its relevant to test at least another, straightforward, stress response of ∆mir-85/ hsp70∆TS mutants to know the specificity. This is important for this study, because the mir-85/hsp70∆TS mutants have apparently no detectable phenotype in normal conditions.

Minor comment:

• What about hs response of alg-2, or some genetic interaction of alg-1 and alg-2?

• "Down-regulation of hsp-70 after heat shock is important for survival". Is the presumed misexpression of Hsp70 in mir/3'UTR mutants required in all tissues to cause the defect, or is there any evidence for tissue-specific preference of regulation for survival. I think the detailed analysis can be a future study, but I wonder if they can comment for discussion eg of mir85 expression or hsp70 induction preferences, since I think this is an important point for considering the biology of this regulation.

Reviewer #4: In this manuscript, Pagliuso and colleagues report on the involvement of microRNA miR85 in the heat-shock recovery (HSR) pathway in C. elegans. Leveraging CRISPR-Cas9 genome engineering and an acute protein degradation method (Auxin Induced Degradation), they show that deletion of ALG-1, miR85, or miR85 binding sites in hsp-70 mRNA are deleterious to the viability of the animal following heat-shock. They identify differentially regulated mRNAs in the miR85 deletion, and they show that loss of miR85, or deletion of miR85-binding sites in hsp-70 mRNA 3’UTR lead to persistence of hsp-70 mRNA and HSP-70 protein in HSR. Lastly, they show that hsp-70 RNAi in HSR rescues the viability phenotype associated with deletion of miR85 or its binding sites.

Overall, this is a sharply focused manuscript with well-controlled experiments and convincing results. The authors unveil the novel linkage of a microRNA and HSR, and convincingly demonstrate the physiological importance of miRNA-mediated repression and decay of the hsp-70 mRNA in this process. The manuscript also features elegant experimental designs including a clever use of AID, and CRISPR edition of 3’UTR sites, which phenocopies miR85 deletion and the HSR viability defects of alg-1 alleles. Considering the importance of HS and the limited prior knowledge on HSR molecular mechanisms, the manuscript should be appealing to a broad readership.

As it is often the case with briefer but exciting papers, the presented results raise several important questions that are left unanswered. Questions such as the reasons for the HSR viability defects resulting from hsp-70 persistence, the nature and mechanism of regulation of the miRNA function during HS remain open, are thoughtfully speculated upon in the Discussion section.

To this reviewer, the most surprising but under-developed piece of data in the manuscript is the observation that hsp-70 mRNA is under limited control by miR85 under normal (non-HS) conditions -hsp-70 mRNA is virtually not de-repressed in hsp-70 delta TS in Figure 3-, and that in spite of not being over-expressed upon HS, miR85 appears to become extremely efficient in the clearance of a strongly induced hsp-70 mRNA. While we recognize that this may not be within the current scope of the manuscript, we suggest two possible explanations that would be worth considering:

1- Is hsp-70 mRNA 3’UTR the same at basal level, and upon HS induction? Alternative polyadenylation sites can lead to dramatically different miRNA responses, and APA usage can be affected by transcription speed and activity. At the very least, the authors could consider performing a 3’RACE on hsp-70 mRNA at basal (non-HS) and upon HS and confirm that the 3’UTR really is what they expect it to be.

2- Another possible mechanism for this is the reorganization of 3’UTR structures upon HS. One could imagine miR85 target site availability to change on hsp-70 mRNA upon ‘melting’ the 3’UTR structures. Can the authors detect candidate folding structures surrounding or overlapping the two neighbouring miR85 binding sites?

Minor comments:

Figure 3C is currently under-developed to an extent that it is next to useless. A more extensive description of the deg, even if only a few in this figure, would improve the value of this interesting experiment.

This is the case for Figure 2A as well, where the authors could include an overlap of the OE mRNAs with predicted miR-85 binding sites beyond hsp-70 mRNA would add value to the manuscript.\\

A consideration of the expression domain of miR85 would be valuable as well (where is it expressed? All cells in C. e. Involve miR85 in hsp70?).

**Have all data underlying the figures and results presented in the manuscript been provided?**

Reviewer #1: None

Reviewer #2: **No: **I couldn't locate the dataset for RNA-Seq experiments.

Reviewer #3: Yes

Reviewer #4: Yes

PLOS authors have the option to publish the peer review history of their article (what does this mean?). If published, this will include your full peer review and any attached files.

Reviewer #1: No

Reviewer #2: No

Reviewer #3: No

Reviewer #4: No

---

## [Decision Letter · Decision Letter 1]

22 Jul 2021

Dear Dr Pasquinelli,

We are pleased to inform you that your manuscript entitled "Recovery from Heat Shock Requires the MicroRNA Pathway in Caenorhabditis elegans" has been editorially accepted for publication in PLOS Genetics. Congratulations!

Yours sincerely,

Eric A Miska, PhD

Associate Editor

PLOS Genetics

Gregory P. Copenhaver

Editor-in-Chief

PLOS Genetics

Comments from the reviewers (if applicable):

Reviewer's Responses to Questions

**Comments to the Authors:**

Reviewer #1: The authors did an excellent job of answering my comments adequately with this revised manuscript. I am therefore supportive of the publication of this interesting study in PLoS Genetics. Before the publication, the authors should confirm the appropriate use of genes nomenclature. For example, I think the microRNA gene name should be italicized mir-85 and not miR-85 (miR corresponds to gene product).

Reviewer #2: Following the revision process authors have addressed all questions previously raised. Authors have added and altered figures to clarify their points and made clarifying changes to the text. They have also added additional information in materials and methods.

Reviewer #4: This reviewer is satisfied by the revisions. The authors thoughtfully answered our queries and even discovered new avenues in the process ( the 3'UTR RNA folding is interesting). We also felt that the many additions to the text and figures in response to the other 3 reviewers were meaningful and improved the manuscript.

**Have all data underlying the figures and results presented in the manuscript been provided?**

Reviewer #1: None

Reviewer #2: Yes

Reviewer #4: Yes

PLOS authors have the option to publish the peer review history of their article (what does this mean?). If published, this will include your full peer review and any attached files.

Reviewer #1: No

Reviewer #2: No

Reviewer #4: No

**Data Deposition**

http://datadryad.org/submit?journalID=pgenetics&manu=PGENETICS-D-21-00466R1

**Press Queries**

---

## [Editor Report · Acceptance letter]

30 Jul 2021

PGENETICS-D-21-00466R1 

Recovery from Heat Shock Requires the MicroRNA Pathway in Caenorhabditis elegans 

Dear Dr Pasquinelli, 

We are pleased to inform you that your manuscript entitled "Recovery from Heat Shock Requires the MicroRNA Pathway in Caenorhabditis elegans" has been formally accepted for publication in PLOS Genetics! Your manuscript is now with our production department and you will be notified of the publication date in due course.

With kind regards,

Zsofi Zombor

PLOS Genetics

On behalf of:
